# Involvement of a Transcription factor, *Nfe2*, in Breast Cancer Metastasis to Bone

**DOI:** 10.3390/cancers12103003

**Published:** 2020-10-16

**Authors:** Di Zhang, Sadahiro Iwabuchi, Tomohisa Baba, Shin-ichi Hashimoto, Naofumi Mukaida, So-ichiro Sasaki

**Affiliations:** 1Division of Molecular Bioregulation, Cancer Research Institute, Kanazawa University, Kakuma-Machi, Kanazawa, Ishikawa 920-1192, Japan; zddione@stu.kanazawa-u.ac.jp (D.Z.); sergenti@staff.kanazawa-u.ac.jp (T.B.); mukaida@staff.kanazawa-u.ac.jp (N.M.); 2Department of Molecular Pathophysiology, Institute of Advanced Medicine, Wakayama Medical University, 811-1 Kimiidera, Wakayama 641-8509, Japan; iwabuchi@wakayama-med.ac.jp (S.I.); hashimot@wakayama-med.ac.jp (S.-i.H.)

**Keywords:** bone metastasis, breast cancer, transcription factor, Wnt pathway

## Abstract

**Simple Summary:**

Breast cancer patients are frequently complicated by bone metastasis, which deteriorates their life expectancy. Bone metastasis is treated with the drugs targeting osteoclast activation, which is mostly observed at the metastasis sites. However, these drugs cannot directly inhibit cancer cell growth and therefore, a novel therapeutic strategy is required to impede cancer cell proliferation at bone metastasis sites. Here, we proved that a transcription factor, NFE2, was expressed selectively in breast cancer cells at bone metastasis sites and contributed crucially to their enhanced proliferation therein, by activating Wnt pathway. Thus, NFE2 can be a novel molecular target to treat breast cancer bone metastasis.

**Abstract:**

Patients with triple negative breast cancer (TNBC) is frequently complicated by bone metastasis, which deteriorates the life expectancy of this patient cohort. In order to develop a novel type of therapy for bone metastasis, we established 4T1.3 clone with a high capacity to metastasize to bone after orthotopic injection, from a murine TNBC cell line, 4T1.0. To elucidate the molecular mechanism underlying a high growth ability of 4T1.3 in a bone cavity, we searched for a novel candidate molecule with a focus on a transcription factor whose expression was selectively enhanced in a bone cavity. Comprehensive gene expression analysis detected enhanced *Nfe2* mRNA expression in 4T1.3 grown in a bone cavity, compared with in vitro culture conditions. Moreover, *Nfe2* gene transduction into 4T1.0 cells enhanced their capability to form intraosseous tumors. Moreover, *Nfe2* shRNA treatment reduced tumor formation arising from intraosseous injection of 4T1.3 clone as well as another mouse TNBC-derived TS/A.3 clone with an augmented intraosseous tumor formation ability. Furthermore, NFE2 expression was associated with in vitro growth advantages of these TNBC cell lines under hypoxic condition, which mimics the bone microenvironment, as well as Wnt pathway activation. These observations suggest that NFE2 can potentially contribute to breast cancer cell survival in the bone microenvironment.

## 1. Introduction

Breast cancer is the most prevalent cancer in females with 280,000 new cases annually in the United States alone [1]. Although prognosis has been progressively improved in advanced countries due to early diagnosis and advanced treatment [2], more than 40,000 breast cancer patients succumb annually in the United States [1], mostly due to metastasis. A meta-analysis study revealed that a median of 12% of patients with stage I–III breast cancer developed bone metastases during a median follow-up of 60 months and that more than 50% patients with metastatic breast cancer had bone metastasis [3]. At bone metastasis sites, cancer cells are presumed to interact with the bone microenvironment to cause bone destruction, where osteoclast activation has a crucial role [4]. Thus, bone metastasis is frequently treated with bisphosphonates and/or denosumab, a monoclonal antibody against the receptor activator of NF-κB (RANK) ligand, to inhibit osteoclast activation, thereby controlling bone metastasis [5]. These drugs have disease-modifying effects but are mostly palliative due to their indirect effects on intraosseous cancer cells [5]. It is, therefore, necessary to develop a novel type of agents, which can inhibit the growth of cancer cells in a bone cavity, based on the understanding the molecular and cellular mechanisms underlying bone metastasis.

With the interaction with a myriad of host resident cells, cancer cells metastasize to distant organs through multiple steps: growth at the primary site, intravasation from the primary tumor site, dissemination through systemic circulation, extravasation into the metastatic organ, and growth at the metastatic organ [6]. Moreover, in order to form a metastatic focus, tumor cells should adapt to the microenvironment of a metastatic organ, which exhibits different cell components including immunocompetent cells from the original organ [7]. Intra-cardiac injection of breast cancer cell lines [8] or injection of human breast cancer cell lines into the mammary fat pads of immune-deficient mice [9] was employed as a breast cancer bone metastasis model but could not reproduce the whole steps of bone metastasis. Thus, using a murine triple-negative breast cancer (TNBC) cell line, 4T1.0, we previously established 4T1.3 clone which could metastasize spontaneously to a bone cavity after its orthotropic injection into mammary fat pads of immunocompetent mice [10]. A subsequent analysis revealed that 4T1.3 clone exhibited constitutively enhanced expression of a chemokine, *Ccl4*, and the enhanced capacity of 4T1.3 clone to metastasize to bone was due to neither accelerated growth at its primary site nor augmented migration to bone, but was ascribed to its higher ability to grow in the bone microenvironment [10].

RNAseq was conducted to elucidate the molecular mechanisms underlying the enhanced capability of 4T1.3 clone to grow in a bone cavity and, as a consequence, revealed the changes in the expression of a wide variety of genes in intraosseously growing 4T1.3 clone, even when compared with intraosseously growing 4T1.0. Inspired by the potential involvement of several transcription factors such as *Runx-2* [11] and *Bach1* in breast cancer bone metastasis [12], we searched for an additional transcription factor(s), which is associated with augmented proliferation of 4T1.3 clone in a bone cavity. We finally provided evidence to indicate that *Nfe2*, a transcription factor involved in normal and malignant hematopoiesis [13,14,15,16,17], could provide breast cancer cells with an advantage to grow efficiently in a bone cavity, thereby accelerating breast cancer bone metastasis.

## 2. Results

### 2.1. Transcription Factor Expression in 4T1.3 Grown in a Bone Cavity

We previously observed that 4T1.3 cells exhibited a higher growth capacity compared with 4T1.0 cells [10]. Consistently, intraosseous injection of 4T1.3 cells formed a larger tumor mass with enhanced cell proliferation, as evidenced by increased Ki-67-positive cell numbers, compared with that of 4T1.0 cells (Figure 1a,b). Moreover, intraosseous injection of 4T1.0 and 4T1.3 cells gave rise to similar CD31-positive areas and VEGF expression in intraosseous tumor sites (Figure 1c,d). Intraosseous injection of either 4T1.0 or 4T1.3 increased intraosseous CD51-positive osteoclast numbers to similar extents, compared with untreated mice (Figure 1e). Consistently, the injection of either clone resulted in similar mRNA expression levels of *Rank*, *Rank ligand* (*Rankl*), and osteoprotegerin (*Opg*) (Figure 1f), which are presumed to be involved in osteoclastogenesis. These observations favor the notion that a higher intraosseous growth capacity of 4T1.3 cells can be ascribed to their intrinsic enhanced growth capacity but neither to augmented neovascularization nor to exaggerated osteoclast activation. Hence, we focused on a transcription factor(s) which can confer intraosseous growth ability on 4T1.3 cells, and searched for a candidate transcription factor(s) that fulfilled all three conditions (Figure 2a) and narrowed down to 13 transcription factor genes (Figure 2b). A subsequent qRT-PCR analysis detected significantly augmented expression of three transcription factors, *Lmo2*, *Myb* and *Nfe2*, in 4T1.3 grown in a bone cavity than in that under in vitro culture conditions, among these 13 transcription factors (Figure 2c). Consistently, immunohistochemical (IHC) analysis detected these three transcription factor proteins in the intraosseous tumor foci arising from intraosseous injection of 4T1.3 clone (Figure 2d).

### 2.2. Enhanced Intraosseous Growth by Nfe2-Expressing Breast Cancer Cells

We next examined the effects of the gene transduction of *Lmo2*, *Myb*, or *Nfe2* on intraosseous growth of parental 4T1.0 cells, which grow less efficiently in a bone cavity compared with 4T1.3 [10]. The transfection with individual transcription factors resulted in the establishment of clones, which exhibited enhanced expression of the corresponding transcription factors as evidenced by qRT-PCR (Figure 3a). When these clones were injected directly into a bone cavity, only the *Nfe2*-expressing clone formed significantly larger tumor foci compared with a control vector-transfected one (Figure 3b) together with enhanced NFE2 protein expression at tumor sites (Figure 3c). However, *Nfe2* transfection failed to change in vitro mRNA expression of *Rank*, *Rankl*, or *Opg* (Figure 3d). Consistently, NFE2 proteins were detected at tumor sites arising from intraosseous injection of 4T1.3 cells but not those of 4T1.0 cells (Figure 3e). On the contrary, no differences were observed on the growth rates between *Nfe2*-expressing and control 4T1.0 clones when they were injected into mouse mammary fat pads (Figure 3f). Moreover, *Nfe2* mRNA was expressed to a larger extent in 4T1.3 obtained from intraosseous tumor sites but not mammary fat pad sites (Figure 3g). These observations would indicate that NFE2 could confer an advantage to grow only in a bone cavity but not at the primary sites, mammary fat pads. In order to evaluate the clinical relevance of *Nfe2* expression, we evaluated the data on overall survival of breast cancer patients. Patients with a higher expression of *Nfe2* exhibited an unfavorable prognosis (Figure 3h), suggesting the clinical significance of *Nfe2* expression among breast cancer patients.

### 2.3. Reduced Intraosseous Growth of Breast Cancer Cells by Nfe2 Expression Ablation

We next examined the effects of *Nfe2* gene expression ablation on intraosseous growth. Additionally, from another mouse TNBC cell line, TS/A, we established TS/A.3 clone, which exhibited a higher capacity to grow in a bone cavity but not at mammary fat pads, compared with its parental clone (Figure 4a,b). *Nfe2* shRNA reduced its gene expression in 4T1.3 and TSA.3 clones (Figure 4c) and protein expression in 4T1.3 (Figure 4d) with few effects on in vitro mRNA expression levels of *Rank*, *Rankl*, and *Opg* (Figure 4e). Concomitantly, *Nfe2* shRNA transduction significantly reduced the tumor growth of 4T1.3 and TS/A.3 clones upon their intraosseous injection (Figure 4f) with few effects on their tumor formation upon their injection into mammary fat pads (Figure 4g). These observations would indicate that NFE2 provided breast cancer cells with a growth advantage in a bone cavity but not at primary sites.

### 2.4. Contribution of NFE2 to Breast Cancer Cell Survival under Hypoxic and Anchorage-Independent Conditions

In order to delineate the molecular mechanism underlying NFE2-mediated enhanced intraosseous tumor formation, we examined the effects of NFE2 expression on in vitro proliferation. Reduced *Nfe2* expression did not impair the in vitro proliferation and survival of 4T1.3 or TS/A.3 clone under normoxic and anchorage-dependent conditions (Figure 5a,c). On the contrary, *Nfe2* shRNA treatment reduced the growth advantage of 4T1.3 or TS/A.3 clone under hypoxic and anchorage-independent conditions (Figure 5b,d), the conditions which resemble the bone microenvironment. Consistently, *Nfe2* gene transduction provided 4T1.0 clone with a growth advantage under hypoxic and anchorage-independent conditions but not under normoxic and anchorage-dependent ones (Figure 5e,f). These observations would indicate that NFE2 could enhance the growth of breast cancer cells in a bone cavity where hypoxia predominates [18]. However, hypoxia-inducible factor *(Hif)1α* expression was not augmented in 4T1.3 cells grown in a bone cavity, compared with those under normoxic in vitro culture conditions (Figure 5g). Moreover, *Nfe2* gene transduction into 4T1.0 cells did not increase the expression level of hypoxia-related genes including *Hif1α*, *Hif2α*, *Myc*, and *Nrf2* compared with a control clone even under hypoxic conditions (Figure 5h,i). Thus, NFE2-mediated growth enhancement may not be ascribed to the enhanced expression of these hypoxia-related transcription factors.

### 2.5. Wnt Pathway Activation in Nfe2-Mediated Enhanced Intraosseous Tumor Formation

No evidence to indicate the direct effects of NFE2 on breast cancer cell proliferation [14,17], prompted us to conduct gene set enrichment analysis (GSEA) to delineate biological pathways and processes that are associated with NFE2-mediated enhanced intraosseous tumor formation. The analysis revealed the enrichment in Wnt pathway in *Nfe2*-expressing 4T1.0, compared with its control clone (Figure 6a). Then, we validated this result by using qRT-PCR. *Nfe2*-expressing 4T1.0 clone displayed enhanced gene expression of Wnt-related molecules including *Axin2* and *Lef1* compared with the control 4T1.0 clone (Figure 6b). Similarly, the expression of Wnt-related molecules, such as *Axin2*, *Lbh*, and *Lef1*, was augmented in intraosseously growing 4T1.3 and TS/A.3 cells, compared with intraosseously growing 4T1.0 and TS/A cells, respectively (Figure 6c,d). We finally examined the involvement of the Wnt pathway in NFE2-mediated growth promotion by using a specific Wnt/β-catenin inhibitor, LF3. The inhibitor did abrogate enhancement in cell proliferation of *Nfe2*-expressing 4T1.0 cells under hypoxic and anchorage-independent conditions (Figure 6e,f). Altogether, NFE2 expression can activate the Wnt pathway, which can confer a growth advantage on tumor cells when they grow in a bone cavity with a low oxygen content.

## 3. Discussion

Like metastasis to other organs, bone metastasis is based on a multi-step process from tumor growth at primary sites to tumor formation in bone [19]. Accumulating evidence indicates the involvement of several transcription factors, *Runx-2* [11] and *Bach1* [12], in breast cancer metastasis to bone. RUNX-2 can promote breast cancer bone metastasis by increasing integrin α5-mediated colonization [20] or inducing RANKL-mediated osteoclast activation [21]. *Bach1* can promote bone metastasis by enhancing the expression of metalloproteinase 1 and a chemokine receptor, CXCR4 [12]. On the contrary, we previously demonstrated that a higher capacity of 4T1.3 clone to metastasize to bone can be ascribed to its higher ability to grow in a bone cavity but neither to enhancement in its colonization, induction of osteoclast activation, nor *Cxcr4* expression [10]. Hence, we presumed that an additional transcription factor(s) may contribute to 4T1.3 clone metastasis to bone by binding cis-elements of *Ccl4* gene, which had an indispensable role in bone metastasis and exhibited enhanced expression [10]. Through combined dry bioinformatics and wet analyses, *Nfe2* has finally been identified as a transcription factor, which can provide breast cancer cells with an intraosseous growth advantage.

NFE2 was originally discovered as a DNA binding protein present in extracts of erythroid cells [22] and was subsequently characterized as a hematopoietic-specific basic-leucine zipper protein which dimerizes with the small Maf subunits [23], thereby binding activator protein (AP)-1-like recognition site [24]. NFE2 was initially presumed to be an indispensable transcription factor for correct regulation of porphobilinogen deaminase (PBGD) gene [25], coding for the third enzyme of the heme biosynthesis. Moreover, NFE2 can regulate α- and β-globin gene transcription [26,27]. Neonates exhibit severe anemia probably compounded by concomitant bleeding due to severe thrombocytopenia, but surviving adults exhibit only mild anemia with compensatory reticulocytosis and splenomegaly [28]. Thus, NFE2 deficiency may be compensated by other transcription factors involved in erythroid genesis and globin gene expression. However, NFE2-deficient mice develop severe thrombocytopenia arising from the intrinsic defects in the megakaryocyte lineage [29], indicating its indispensable role in megakaryocyte biogenesis. In contrast, mice expressing an *Nfe2* transgene in hematopoietic cells exhibit many features of myeloproliferative disorders (MPN) including thrombocytosis and leukocytosis, and often develop acute myeloid leukemia [14] or myelosarcoma [17]. Consistently, MPN patients display *Nfe2* overexpression arising from either recurrent truncating mutation [15] or epigenetic changes mediated by the histone demethylase JMJD1C [16]. These observations would indicate the crucial roles of aberrant *Nfe2* expression in some types of leukemogenesis.

NFE2 expression is presumed to be restricted to hematopoietic cells such as erythroid cells, megakaryocytes and granulocytes [30,31]. On the contrary, we provided the first definitive evidence to indicate that NFE2 was robustly expressed in mouse breast cancer cells grown in a bone cavity. Moreover, we proved that NFE2 could confer a growth advantage on breast cancer cells under hypoxic and anchorage-independent conditions, which mimic the microenvironment in a bone cavity. The studies on *Nfe2* transgenic mice proved that *Nfe2* gene overexpression could increase colony formation of several types of hematopoietic progenitor cells, but without additional elucidation on its molecular mechanisms [14,17]. Hence, we conducted GSEA on *Nfe2*-overexpression 4T1.0 clone and demonstrated, together with subsequent qRT-PCR analysis, the enrichment of Wnt pathway molecules in 4T1.3 clones grown in a bone cavity as well as *Nfe2*-expressing breast cancer cells. The Wnt pathway is involved in various aspects of carcinogenesis, such as cell proliferation, resistance to cell death, and angiogenesis [32]. A Wnt inhibitor abrogated NFE2-mediated enhancement in cell proliferation and survival observed on *Nfe2*-overexpressing cells which were cultured under hypoxic conditions. Given that bone cavity is generally hypoxic, it is likely that the activated Wnt pathway is associated with enhanced breast cancer cell survival in the bone cavity.

NF-E2-related factor (Nrf2) is a master regulatory transcription factor of cytoprotective genes [33], and its aberrant expression is frequently observed in cancer cells and can provide an advantage for their growth [34]. Nrf2 is also a basic-leucine zipper DNA-binding protein and shows a structural similarity with Nfe2 even at its DNA-binding domain [35]. Like NFE2, Nrf2 heterodimerizes with the small Maf subunit [36], and binds to anti-oxidant responsive element (ARE) or electrophile-responsive element (EpRE) to induce transcription of target genes [37]. Moreover, several lines of evidence indicate that aberrant Nrf2 expression could activate Wnt pathways in several types of carcinogenesis [38,39]. The consensus ARE/EpRE sequence comprises 5′-TGACNNNGC-3′ [37], which is highly similar to the consensus-binding sequence for *Nfe2* [40]. Hence, it is tempting to speculate that NFE2 might activate similar sets of target genes as Nrf2 did and eventually activate the Wnt pathway.

Analyses on the public database, GeneCards^®^ (https://www.genecards.org), together with our RNAseq on intraosseously grown 4T1.3 and 4T1.0 cells, demonstrated that NFE2 can regulate the expression of several bone metastasis-related transcription factors including *Runx2* [11], *Bach1* [12], and Esr1 [41]. Thus, NFE2 may be a hub of the transcription factors involved in bone metastasis and consequently its blockade may be effective to treat bone metastasis. NFE2 and Nrf2 bind with a similar region of the small Maf subunit to form a heterodimer and to eventually exert their functions [13,33]. As ML385, a thiazole-indoline compound, can specifically inhibit the interaction between Nrf2 and Maf [42], it may be able to inhibit the interaction between NFE2 and Maf, thereby blocking NFE2 activities. Alternatively, inhibition of *Nfe2* expression can be employed to treat bone metastasis of breast cancer with aberrant *Nfe2* expression. As reported on *Runx-2* [43], the treatment with siRNA may be feasible to decrease *Nfe2* expression. Moreover, *Nfe2* transcription could be enhanced by another transcription factor, *Runx-1* [44]. Thus, *Nfe2* expression can be reduced by the modulation of the expression and/or functions of this transcription factor. Furthermore, *Nfe2* expression in hematopoietic cells can be enhanced also by several secreted bioactive molecules such as activin A [45] and platelet-derived growth factor (PDGF) [46]. As our preliminary observations demonstrated that *Pdgf* mRNA expression was increased in fibroblasts present in 4T1.3-growing bone cavity, fibroblast-derived PDGF may enhance *Nfe2* expression in bone metastasis foci. Hence, PDGF blockade may be also effective to treat bone metastasis by decreasing *Nfe2* expression. Nevertheless, more detailed analysis on the functions and expression mechanisms of NFE2 will be warranted to establish a successful strategy to treat bone metastasis by targeting NFE2.

## 4. Materials and Methods

### 4.1. Cell Lines

A mouse TNBC cell line, BALB/c-derived 4T1 (CRL-2539), was purchased in 2007 from American Type Culture Collection. Two clones, parental 4T1.0 clone and 4T1.3 clone with a high capacity to metastasize to bone, were established from 4T1 as previously described [10]. TS/A, another mouse TNBC cell line established from BALB/c mammary adenocarcinoma [47], was obtained in 2010 from Dr. Nanni. To establish a TS/A clone with a high growth capacity in the bone marrow cavity, parental TS/A cells (2.5 × 10^3^ cells in 20 µL HBSS) were injected into a bone marrow cavity of BALB/c mouse tibiae. Eleven days after the tumor inoculation, total bone marrow cells were collected and cultured for 3 weeks in medium containing 50–100 µg/mL G418. The resultant cells, TS/A.1 clone, were injected again into a bone marrow cavity of a new mouse tibiae, and this cycle was repeated three times to obtain TS/A.3 clone. All these cell lines were cultured at 37 °C under 5% CO_2_ in a RPMI 1640 medium supplemented with 10% fetal bovine serum (FBS) and used within 10 passages of receiving from the source or the establishment of the clone. 

### 4.2. Mice

Eight-week old specific pathogen-free BALB/c mice were obtained from Charles River Laboratories (Yokohama, Japan) and were kept under the specific pathogen-free conditions. All the animal experiments in this study complied with the Guideline for the Care and Use of Laboratory Animals of Kanazawa University with the approval by the Committee on Animal Experimentation of Kanazawa University (AP-163706).

### 4.3. Antibodies

Mouse anti-mouse pan cytokeratin antibody (BioLegend, San Diego, CA, USA), rabbit anti-mouse LMO2 and rabbit anti-mouse NFE2 antibodies (Novus Biologicals, Centennial CO, USA), rabbit anti-mouse c-MYB antibody (Bioss Antibodies, Woburn, MA, USA), rat anti-mouse Ki-67 antibody (Dako Japan, Kyoto, Japan), mouse anti-human CD51 antibody (BD Biosciences, San Diego, CA, USA), rabbit anti-mouse CD31 antibody (Abcam, Toronto, ON, Canada), and goat anti-mouse VEGF antibody (Santa Cruz, Sana Cruz, CA, USA) were used as the primary antibodies for immunohistochemical analysis. The following antibodies were used as the primary antibodies for flow cytometry analysis; Allophycorin (APC)-labeled rat anti-mouse CD326 and peridinin–chlorophyll–protein complex–cyanine 5.5 (PerCP/Cy5.5)-labeled rat anti-mouse CD24 antibodies (BioLegend), FITC- or PerCP/Cy5.5-labeled rat anti-mouse CD45 antibodies (eBioscience, San Diego, CA, USA), APC-labeled rat-anti mouse CD44 antibody (BD Biosciences).

### 4.4. RNAseq Analysis

Total RNAs were extracted from the cell lines and its quality was confirmed using an Agilent 2200 TapeStation (Agilent Technologies, Palo Alto, CA, USA). RNA Integrity Numbers (RIN) of all samples were 10.0, and were subjected to RNAseq analysis as previously described. GSEA was executed through the GenePattern server [48,49,50]. As the metric for ranking genes, log2_ratio_of_means was exploited, and the gene set size was set to 20 to 200.

### 4.5. Procedures to Identify Candidate Transcription Factors

Candidate transcription factors were selected as those that fulfilled all three of the following conditions (Figure 2a). The first condition is that the candidate transcription factor gene expression should be confirmed to be more abundant in 4T1.3 cells grown in a bone cavity than 4T1.0 cells grown in a bone cavity with the use of RNAseq analysis on tumor foci arising from intraosseous injection of 4T1.0 or 4T1.3 cells [10]. The second condition is that the binding site of the candidate transcription factors should be verified to be localized in the region from 2 kbp upstream to 200 bp downstream of the transcription start site of mouse *Ccl4* gene, which is abundantly expressed in 4T1.3 cells [10], with the use of TFBIND (http://tfbind.hgc.jp) at a cut-off value higher than 0.85. The third condition is that the candidate transcription factor gene should be proven to be expressed more abundantly in bone metastasis sites than lung metastasis sites in GSE14020, which is a dataset of genes expressed in various metastatic sites of breast cancer patients.

### 4.6. Establishment of Cell Lines Constitutively-Expressing Transcription Factors

Full length construct of mouse *Lmo2* (MMM1013-202769386), *Myb* (MMM1013-202763262) and *Nfe2* cDNAs (MMM1013-202804796) were purchased from Dharmacon (GE Healthcare Dharmacon, Milwaukee, WI, USA). Each protein coding portion is cloned into the pIRES vector (Takara Bio USA, Mountain View, CA, USA, PT3266-5) using suitable restriction sites. The resultant expression vectors were transduced into 4T1.0 clone using JetPRIME DNA and siRNA transfection reagent (Polyplus transfection, Illkirch, France). Then, the cells were cultured in medium containing G418 (400 µg/mL) for 7 days, to obtain constitutively expressing clones.

### 4.7. Establishment of Cell Lines Expressing shRNA

MISSION® shRNA Lentiviral Transduction Particles (Sigma Aldrich, St. Louis, MO, USA, SHCLNV-NM_008685) were used to knockdown the *Nfe2* gene expression in the 4T1.3 and TS/A.3 clones. shRNAs were generated against the mouse *Nfe2* mRNA (TRC Number: TRCN0000081983 and TRCN0000424174) and were packaged into the lentiviral particles. shRNAs targeting scrambled RNAs (Nontarget; SHC002) were prepared with the use of pLK01-puro (Sigma-Aldrich, St. Louis, MO, USA) according to the manufacturer’s instructions. Stably gene knocked-down clones were selected in the presence of puromycin (3.5 µg/mL for 4T1.3 cell line and 6 µg/mL for TS/A.3 cell line).

### 4.8. Intraosseous Injection

Tumor cell suspensions (5.0 × 10^3^ 4T1.0-derived cells or 10 × 10^3^ TS/A-derived ones in 20 µL HBSS) were injected into a bone marrow cavity of tibiae as previously described [51]. Seven or nine days after the injection, tibiae were removed for a subsequent analysis. In order to delineate transcription factor expression, the resected tibiae were subjected to the immunostaining with anti-LMO2, -MYB or -NFE2 antibodies by using the ABC Elite Kit (Vector Laboratories, Burlingame, CA, USA) and peroxidase substrate 3, 3′-diaminobenzidine kit (Vector Laboratories) in an essentially similar way as described previously [10]. To evaluate intraosseous tumor mass arising from 4T1.0-derived clones, the resected mouse tibiae were immunostained with anti-pan cytokeratin antibodies and the ratios of pan-cytokeratin-positive to whole bone marrow areas were determined with the use of a microscopy system (BZ-X700) and Keyence Analysis Software (Keyence Japan, Osaka, Japan), as previously described [10]. To evaluate osteoclast numbers in a bone cavity, resected mice tibiae were subjected to the immunostaining with anti-CD51 antibody, and the ratios of CD51-positive areas to the whole bone marrow areas were determined. To evaluate tumor cell proliferation and angiogenesis in a bone cavity, resected mouse tibiae were subjected to immunostaining with anti-Ki-67, -CD31 or -VEGF antibodies, and the ratios of each of the positive cells or positive areas to the whole tumor foci areas were determined. For the enumeration of tumor cell numbers in bone marrow, single cell suspensions were prepared from the resected tibial bones, by sequentially flushing with the RPMI medium and removing erythrocytes with ammonium chloride lysis buffer. The resulting single cell suspensions were incubated with either the combination of FITC-labeled anti-CD45, PerCP/Cy5.5-labeled anti-CD24, and APC-labeled anti-CD44 antibodies, or the combination of PerCP/Cy5.5-labeled anti-CD45 and APC-labeled anti-CD326 antibodies, for 30 min at 4 °C. Dead cells were removed from acquired data with a fixable viability Dye (eBioscience). The stained cells were acquired on a FACS Canto System II (BD Biosciences) and analyzed using FlowJo software (BD Biosciences). 4T1.0-derived and TS/A-derived tumor cells were defined as CD45^−^CD326^+^ and CD45^−^CD24^−^CD44^high^ cells, respectively. In another series of experiments, 4T1.0-derived tumor cells were isolated and collected by using FACS Aria cell sorter (BD Biosciences) and subjected to RNA extraction immediately.

### 4.9. In Vitro Cell Proliferation Assay

The cell suspensions (2.0 × 10^5^ cells in 10 mL) were seeded in 100 mm dish and cultured at 37 °C for the indicated time intervals. Live cell numbers were determined by using trypan blue exclusion assay while survival rates were calculated as the ratios of live cell numbers to the sum of live and dead cells. In some experiments, cell suspensions (2.0 × 10^5^ cells in 500 µL) were added to each well of an EZ-BindShut^®^ 24-well plate (IWAKI Glass, Chiba, Japan) and incubated at 37 °C under hypoxic (1% O_2_) and anchorage-independent conditions for the indicated time points to determine live cell numbers and survival rates. In another series of experiments, Nfe2-expressing or control vector-transfected 4T1.0 cells were seeded and cultured in normoxic (20% O_2_) and anchorage-dependent, or hypoxic and anchorage-independent conditions in the presence of Wnt inhibitor, LF3 (30 μM, SelleckChem, Houston, TX, USA) or DMSO to determine live cell numbers and survival rates. In these experiments, culture media were not changed until the end of the experiments.

### 4.10. Tumor Growth at the Primary Site

The cell suspensions (1.5 × 10^5^ cells in 100 µL) were injected into the secondary mammary fat pads and tumor growth was evaluated by measurement with calipers every 2 to 3 days. Tumor volumes were calculated as *a* × *b*^2^/2, where *a* and *b* mean the long and the short diameters of the tumor, respectively. To collect tumor cells from primary tumor tissues, single cell suspensions were prepared from the resected primary tumors 9 days after orthotopic injection. Primary tumors were cut into small pieces and treated with 1 mg/mL collagenase D (Roche, Basle, Switzerland) and 40 μg/mL bovine pancreas-derived deoxyribonuclease I (Sigma-Aldrich) for 20 min and 37 °C with constant shaking. The resulting mixtures were filtered through a 100 μm cell strainer (Corning, Corning, NY, USA) to obtain single cell suspensions. Erythrocytes were removed with ammonium chloride lysis buffer. The resulting single cell suspensions were incubated with the combination of FITC-labeled anti-CD45, PerCP/Cy5.5-labeled anti-CD24, PE-labeled anti-CD44, and APC-labeled anti-CD326 antibodies, for 30 min at 4 °C. Dead cells were removed from acquired data with a fixable viability Dye (eBioscience). Tumor cells were defined as CD45^−^CD326^+^CD24^−^CD44^high^ cells and were collected by using a FACS Aria cell sorter (BD Biosciences). Isolated tumor cells were subjected to total RNAs extraction immediately.

### 4.11. qRT-PCR Analysis

Total RNAs were extracted from cell lines or tumors with the use of RNeasy Mini Kit (Qiagen, Chatsworth, CA, USA) and subjected to qRT-PCR by using the primers listed in Appendix A
Table A1, as described previously [52]. Expression levels of the target genes were analyzed through the comparative threshold cycle method (ΔΔCT). The hypoxanthine guanine phosphoribosyltransferase (*Hprt*) were used as an internal control.

### 4.12. Clinical Database Analysis

Overall survival (GSE16446) was evaluated by analyzing the data deposited in the Kaplan–Meier Plotter database (https://kmplot.com/analysis/) and was shown using Kaplan–Meier curve. The cohort was divided by using the cutoff value 6 in analysis to set the patients into 2 groups, high and low NFE2-expressing groups.

### 4.13. Library Construction and Sequencing

Total RNAs were obtained from the in vitro cultured parental or *Nfe2*-expressing 4T1 cells by a RNeasy Plus Mini Kit (Qiagen), following the instructions. The quality of RNAs was evaluated using an Agilent 4200 TapeStation (Agilent Technologies), and the concentration was measured using a NanoDrop (Thermo Fisher Scientific, Waltham, MA, USA). The libraries for sequencing were created using TruSeq Stranded mRNA (Illumina, San Diego, CA, USA), following the procedure, and their quality and quantities were measured by using an Agilent 4200 TapeStation, a Qbit Fluorometer (Thermo Fisher Scientific) and a KAPA Library Quantification kit (Roche). The average size of the libraries was 292–340 bp, and high-throughput sequencing for 6 samples was performed by NextSeq 500/550 High Output Kit v2.5 (75 cycles, 40/40 cycles, pair-end). All datasets have been deposited at the National Center for Biotechnology Information, Gene Expression Omnibus under accession number 154,541 (https://www.ncbi.nlm.nih.gov/geo/query/acc.cgi?acc=GSE154541). The result of bulk RNA sequencing was calculated by using a CLC Genomics Workbench (Filgen, Inc., Aichi, Japan). Then we investigated c6 (c6.all.v7.1.symbols.gmt) gene sets in the current study. The significant results of GSEA were identified using the criteria: FDR *p*-value < 0.05 and FDR < 0.25.

### 4.14. Statistical Analysis

The means + SD were calculated for all parameters determined. Statistical significance was evaluated using one-way ANOVA, followed by the Tukey–Kramer post-hoc test or the Mann–Whitney *U* test. *p* values of less than 0.05 were considered statistically significant.

## 5. Conclusions

In this study, we revealed that a transcription factor, *Nfe2*, plays an indispensable role in breast cancer metastasis to bone, by conferring growth advantages to breast cancer cells in the bone microenvironment. Moreover, *Nfe2* expression was associated with enhanced expression of Wnt-related molecules, suggesting that NFE2 can regulate the Wnt pathway, thereby contributing to breast cancer growth in a bone cavity.

## Figures and Tables

**Figure 1 cancers-12-03003-f001:**
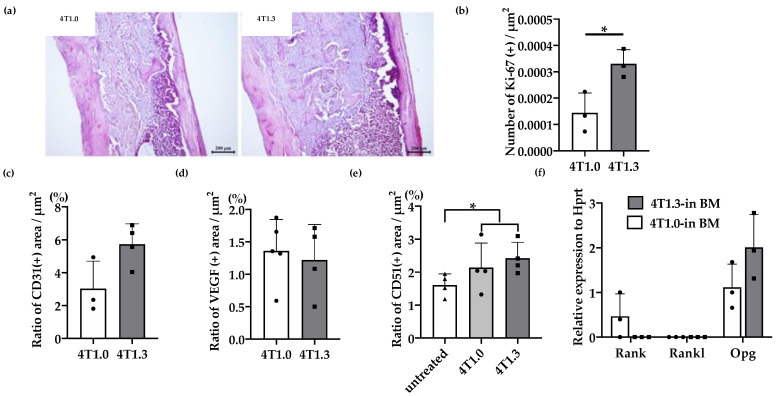
(**a**) H&E staining of 4T1.0 or 4T1.3 tumor foci in a bone cavity. Tibiae were obtained from mice 7 days after intraosseous injection of 4T1.0 or 4T1.3 cells for H&E staining. Representative images from 5 independent mice are shown here. Scar bars, 200 μm. (**b**) Enhanced cell proliferation at intraosseous tumor sites. Tibiae were obtained from mice 7 days after intraosseous injection of 4T1.0 or 4T1.3 cells and were subjected to anti-Ki67-antibody staining. All values represent mean + SD (*n* = 3). * *p* < 0.05. (**c**,**d**) Quantitative analysis of CD31- and VEGF-positive areas in 4T1.0 or 4T1.3 tumor foci in a bone cavity. Tibiae were obtained from mice 7 days after intraosseous injection of 4T1.0 or 4T1.3 cells and were subjected to anti-CD31 (**c**) or anti-VEGF (**d**) antibody staining. All values represent mean + SD (*n* = 3 to 5). (**e**) Quantitative analysis of CD51-positive areas in tumor-bearing bones. Tibiae were obtained from mice untreated or 7 days after intraosseous injection of 4T1.0 or 4T1.3 cells and were subjected to anti-CD51 antibody staining. All values represent mean + SD (*n* = 4). * *p* < 0.05. (**f**) Determination of intraosseous mRNA expression level of *Rank*, *Rankl*, and *Opg* in 4T1.0- or 4T1.3-injected mice. Total RNAs were extracted either from 4T1.0 or 4T1.3 cells isolated from a bone marrow cavity at day 7 after intraosseous injection. All values represent mean + SD (*n* = 3).

**Figure 2 cancers-12-03003-f002:**
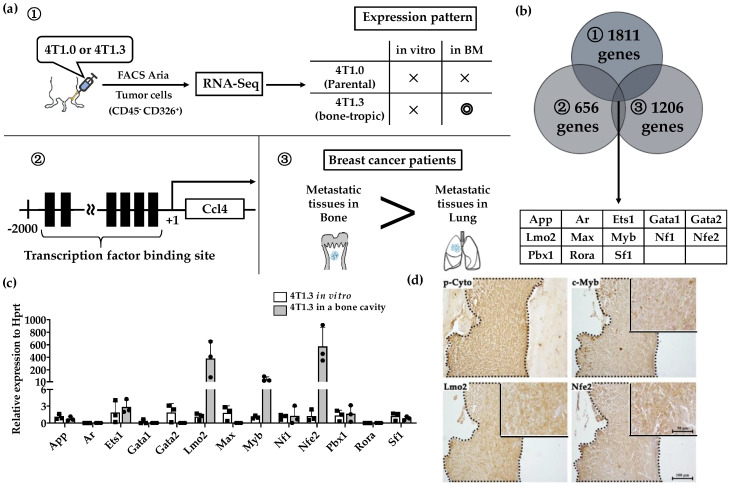
(**a**) Schematic representation of the procedures to identify candidate Transcription factors. (**b**) A total of 13 transcription factors fulfilled 3 conditions described in Figure 2a. (**c**) mRNA expression level of 13 extracted transcription factors in 4T1.3 cells, which grew in a bone marrow cavity or were cultured in vitro. Total RNAs were extracted from 4T1.3 cells, either isolated from the bone marrow cavity at day 7 after intraosseous injection, or cultured in vitro. The expression levels of 13 transcription factors were determined by qRT-PCR. The mean and SD values were calculated from 3 animals or 3 independently in vitro cultured cells. (**d**) Immunohistochemical analysis of 4T1.3 cells in a bone cavity. Tibiae were obtained from mice 7 days after intraosseous injection of 4T1.3 cells and were subjected to IHC staining by using anti-pan cytokeratin, anti-LMO2, anti-MYB, or anti-NFE2 antibodies. The tumor foci were indicated as the positive areas of pan cytokeratin, which were demarcated by dot lines. Representative images from 5 independent mice are shown here. Insets indicate enlarged area of tumor foci. Scar bars, 100 μm in main images, 50 μm in insets.

**Figure 3 cancers-12-03003-f003:**
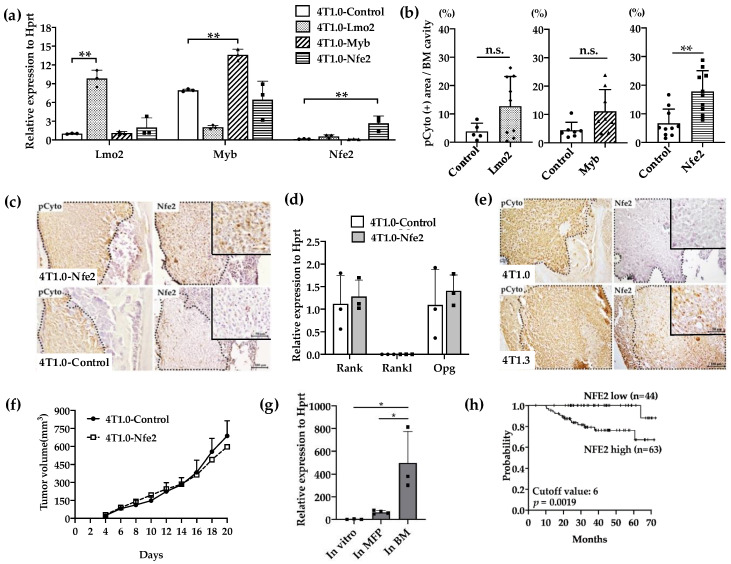
(**a**) Expression levels of transcription factor mRNA expression in the used transfectants. 4T1.0 cells were transfected with each transcription factor-expressing or control vector and total RNAs were extracted from the resultant transfectants. The mean and SD values were calculated from 3 independent experiments. ** *p* < 0.01. (**b**) Intraosseous tumor formation of *Lmo2*-, *Myb*-, or *Nfe2*-overexpressing 4T1.0 cell lines. *Lmo2*-, *Myb*- or *Nfe2*-expressing or control vector-transfected 4T1.0 cells (5.0 × 10^3^ cells) were injected into tibiae of mice. Seven days after the inoculation, tibiae were removed and subjected to anti-pan cytokeratin antibody staining to detect tumor foci formation. All values represent mean + SD (*n* = 5 to 9). ** *p* < 0.01, n.s., not significant. (**c**) Immunohistochemical analysis of *Nfe2*-expressing or control vector-transfected 4T1.0 cells in a bone cavity. Tibiae were obtained from mice 7 days after intraosseous injection of Nfe2-expressing or control vector-transfected 4T1.0 cells and were subjected to anti-pan cytokeratin or anti-NFE2 antibodies staining. Pan-cytokeratin-positive tumor areas are demarcated by dot lines. Representative images from 5 independent mice are shown here. Insets indicate enlarged area of tumor foci. Scar bars, 100 μm in main images, 50 μm in insets. (**d**) Determination of mRNA expression level of m *Rank*, *Rankl*, or *Opg* in *Nfe2*-expressing or control vector-transfected 4T1.0 cells. Total RNAs were extracted either from *Nfe2*-expressing or control vector-transfected 4T1.0 cells cultured in vitro. All values represent mean + SD (*n* = 3). (**e**) Immunohistochemical analysis of 4T1.0 or 4T1.3 cells in a bone cavity. Tibiae were obtained from mice 7 days after intraosseous injection of 4T1.0 or 4T1.3 cells and were subjected to anti-pan cytokeratin or anti-NFE2 antibodies staining. Pan-cytokeratin-positive tumor areas are demarcated by dot lines. Representative images from 5 independent mice are shown here. Insets indicate enlarged area of tumor foci. Scar bars, 100 μm in main images, 50 μm in insets. (**f**) Tumor growth rates of *Nfe2*-expressing or control vector-transfected 4T1.0 cells upon their injection into mammary fat pads. *Nfe2*-expressing or control vector-transfected 4T1.0 cells (1.5 × 10^5^) were injected into mammary fat pads and tumor growth was determined at the indicated time points. All values represent mean + SD (*n* = 5). (**g**) Expression level of *Nfe2* mRNA in 4T1.3 cells, which were cultured in vitro, grew at mammary fat pads or in a bone marrow cavity. Total RNAs were extracted from 4T1.3 cells, which were cultured in vitro (In vitro), were isolated from primary tumor at day 9 after orthotopic injection (In MFP) or from a bone marrow cavity at day 7 after intraosseous injection (In BM). All values represent mean + SD (*n* = 3). * *p* < 0.05 (**h**) Effects of NFE2 expression level on the prognosis of breast cancer patients. Overall survival of patients with breast cancer (GSE16446) was analyzed based on NFE2 expression level by analyzing the Kaplan–Meier Plotter database.

**Figure 4 cancers-12-03003-f004:**
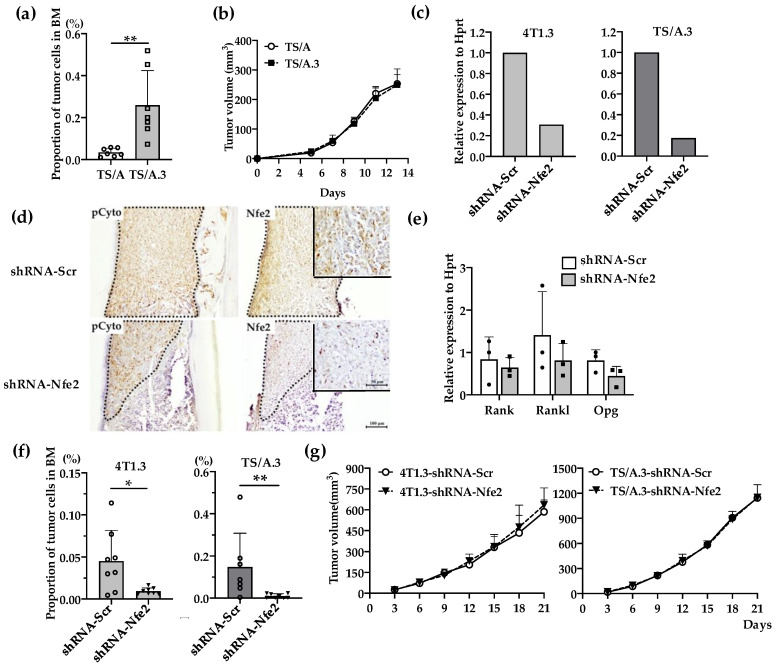
(**a**) Intraosseous injection of TS/A and TS/A.3 clones. Tibia was obtained from mice 9 days after intraosseous injection of TS/A or TS/A.3 cells and was subjected to flow cytometric analysis. The proportions of CD45^−^ CD24^−^ CD44^high^ tumor cells among total cells were determined as described in the Materials and Methods section. All values represent mean + SD (*n* = 7). ** *p* < 0.01. (**b**) Tumor growth rates of TS/A and TS/A.3 cells upon their injection into mammary fat pads. TS/A or TS/A.3 cells (1.5 × 10^5^) were injected into mammary fat pads and tumor growth was determined at the indicated time points. All values represent mean + SD (*n* = 4). (**c**) *Nfe2* expression level in *Nfe2* shRNA-transfected 4T1.3 or TS/A.3 clones. Either 4T1.3 or TS/A.3 clones were transfected with *Nfe2* or scrambled shRNA-expressing vectors. Total RNAs were extracted from the resultant transfectants and were subjected to qRT-PCR to quantify Nfe2 mRNA. (**d**) Immunohistochemical analysis of *Nfe2* shRNA-transfected 4T1.3 cells in a bone cavity. Tibiae were obtained from mice at day 9 after intraosseous injection of *Nfe2* shRNA- or scrambled shRNA-transfected 4T1.3 cells and were subjected anti-pan cytokeratin, or anti-NFE2 antibodies staining. Pan-cytokeratin-positive tumor areas are demarcated by dot lines. Representative images from 5 independent mice are shown here. Insets indicate enlarged area of tumor foci. Scar bars, 100 μm in main images, 50 μm in insets. (**e**) Determination of mRNA expression level of *Rank, Rankl*, or *Opg* in *Nfe2* shRNA-transfected 4T1.3 cells. Total RNAs were extracted either from *Nfe2* shRNA- or scrambled shRNA-transfected 4T1.3 cells cultured in vitro. The mRNA expression levels were determined by qRT-PCR. All values represent mean + SD (*n* = 3). (**f**) Intraosseous tumor formation of *Nfe2* shRNA-transfected 4T1.3 or TS/A.3 clones. Tibiae were obtained from mice 9 days after intraosseous injection of *Nfe2* shRNA- or scrambled shRNA-transfected 4T1.3 or TS/A.3 cells and were subjected to flow cytometric analysis. All values represent mean + SD (*n* = 7 to 8). * *p* < 0.05. (**g**) Tumor growth rates of *Nfe2* shRNA-transfected 4T1.3 or TS/A.3 clones upon their injection into mammary fat pads. *Nfe2* shRNA- or scrambled shRNA-transfected 4T1.3 or TS/A.3 clones (1.5 × 10^5^) were injected into mammary fat pads and tumor growth was determined at the indicated time points. All values represent mean + SD (*n* = 5).

**Figure 5 cancers-12-03003-f005:**
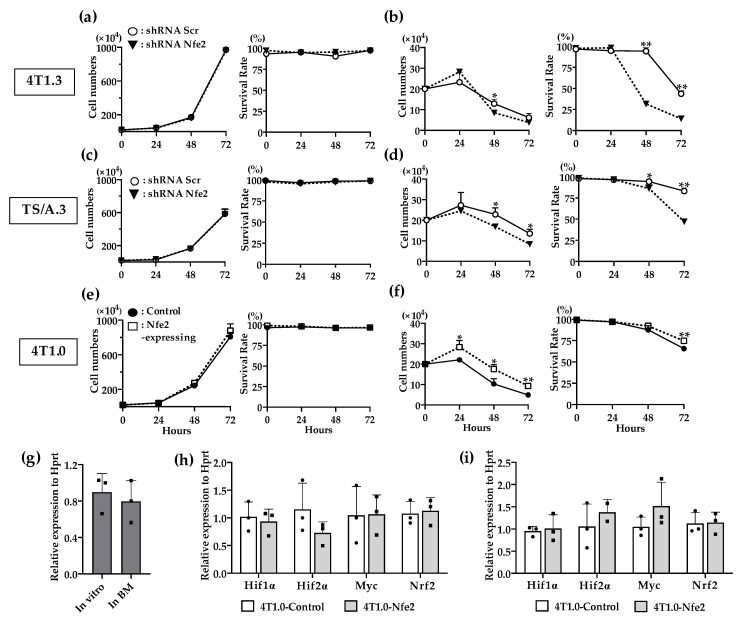
(**a**,**b**) In vitro proliferation rates of *Nfe2* shRNA- or scrambled shRNA-transfected 4T1.3 cells under normoxic (**a**) or hypoxic and anchorage-independent conditions (**b**). (**c**,**d**) In vitro proliferation rates of *Nfe2* shRNA- or scrambled shRNA-transfected TS/A.3 cells under normoxic (**c**) or hypoxic and anchorage-independent conditions (**d**). (**e**,**f**) In vitro proliferation rates of *Nfe2*-expressing or control vector-transfected 4T1.0 cells under normoxic (**e**) or hypoxic and anchorage-independent conditions (**f**). Each of the cell lines were cultured under normoxic (20% O_2_) or hypoxic (1% O_2_) and anchorage-independent conditions, respectively. Cell numbers and survival rates under each condition were determined by using trypan blue exclusion assay. All values represent mean + SD (*n* = 3). * *p* < 0.05, ** *p* < 0.01. (**g**) Expression level of *Hif1α* mRNA expression in 4T1.3 cells, which were cultured in vitro, or grew in a bone cavity. Total RNAs were extracted from 4T1.3 cells, which were cultured in vitro (In vitro) or grew in a bone cavity (In BM) at day 7 after intraosseous injection. All values represent mean + SD (*n* = 3). (**h**,**i**) Determination of mRNA expression level of hypoxia-related genes in *Nfe2*-expressing or control vector-transfected 4T1.0 cells, which was cultured under normoxic or hypoxic and anchorage-independent conditions. Total RNAs were extracted from *Nfe2*-expressing or control vector-transfected 4T1.0 cells cultured under normoxic (**h**) or hypoxic and anchorage-independent conditions (**i**), respectively. All values represent mean + SD (*n* = 3).

**Figure 6 cancers-12-03003-f006:**
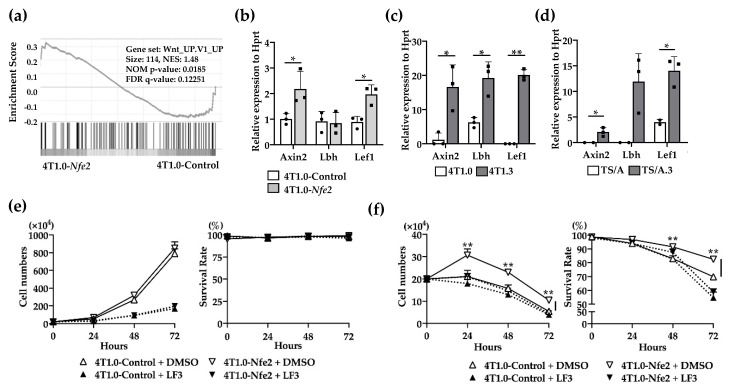
(**a**) Gene set enrichment analysis of Nfe2-expressing or control vector-transfected 4T1.0 cells in vitro. Total RNAs were extracted from *Nfe2*-expressing or control vector-transfected 4T1.0 cells under in vitro culture conditions and were subjected to bulk RNA sequencing. (**b**) Determination of mRNA expression of Wnt-related molecules in *Nfe2*-expressing or control vector-transfected 4T1.0 cells in vitro. Total RNAs were extracted from *Nfe2*-expressing or control vector-transfected 4T1.0 cells under in vitro culture conditions. All values represent mean + SD (*n* = 3). * *p* < 0.05. (**c**) Determination of mRNA expression of Wnt-related molecules in 4T1.0 or 4T1.3 cells, which grew in a bone cavity. Total RNAs were extracted from 4T1.3 cells or its parental 4T1.0 cells, which were isolated from a bone cavity at day 9 after intraosseous injection as described in the Materials and Methods section. Comparative CT values were calculated by normalizing to each value of 4T1.0 cells cultured under in vitro conditions. All values represent mean + SD (*n* = 3). * *p* < 0.05, ** *p* < 0.01. (**d**) Determination of mRNA expression of Wnt-related molecules in TS/A or TS/A.3 cells, which grew in a bone cavity. Total RNAs were extracted from TS/A.3 cells or its parental TS/A cells, which were isolated from a bone cavity at day 9 after intraosseous injection. Comparative CT value were calculated by normalizing to each value of TS/A cells cultured under in vitro conditions. All values represent mean + SD (*n* = 2 to 3). * *p* < 0.05. (**e**,**f**) In vitro proliferation rates of *Nfe2*-expressing or control vector-transfected 4T1.0 cells in the presence of LF3 under normoxic (**e**) or hypoxic and anchorage-independent conditions (**f**). Each of the cell lines were cultured with LF3 or vehicle control, DMSO, under normoxic (20% O_2_) or hypoxic (1% O_2_) and anchorage-independent conditions, respectively. Cell numbers and survival rates were determined by using trypan blue exclusion assay. All values represent mean + SD (*n* = 3). ** *p* < 0.01.

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
