# Peer review of "Involvement of a Transcription factor, Nfe2, in Breast Cancer Metastasis to Bone"

_cancers, 2020, doi:10.3390/cancers12103003_

Round 1

Reviewer 1 Report

Manuscript ID: cancers-960120-R1

            The revised version of the manuscript  entitled” Involvement of a transcription factor, Nfe2, in breast      cancer metastasis to bone” by Dr. Di Zhang et al., present some data using a 4T1.3 tumour cell clone with a high capacity to metastasize to bone after orthotopic injection, derived from a murine TNBC cell line, 4T1.0.They describe some aspects of the molecular mechanism underlying a high growth ability of 4T1.3 in a bone cavity. They performed a gene expression analysis and they detected enhanced Nfe2 mRNA expression in 4T1.3 grown in a bone cavity, compared with in vitro culture conditions. They showed that Nfe2 gene transduction into 4T1.0 cells enhanced their capability to form intraosseous tumours, whereas Nfe2 shRNA treatment reduced tumour formation of 4T1.3 clone, as well as another mouse TNBC-derived TS/A.3 clone with an augmented intraosseous tumour formation ability. They propose that Nfe2 can potentially contribute to breast cancer cell survival in bone microenvironment under hypoxic condition.

            The manuscript has been improved by incorporating some of the suggested experiments. Although not all the suggestions have been incorporated, I believe that the data are more consistent and better supported than the previous version.

In summary the addition of new experiments and clarifications in the text have improved the final manuscript. In my opinion, it is almost acceptable for publication.

Author Response

Reviewer #1.

The revised version of the manuscript  entitled” Involvement of a transcription factor, Nfe2, in breast cancer metastasis to bone” by Dr. Di Zhang et al., present some data using a 4T1.3 tumour cell clone with a high capacity to metastasize to bone after orthotopic injection, derived from a murine TNBC cell line, 4T1.0.They describe some aspects of the molecular mechanism underlying a high growth ability of 4T1.3 in a bone cavity. They performed a gene expression analysis and they detected enhanced Nfe2 mRNA expression in 4T1.3 grown in a bone cavity, compared with in vitro culture conditions. They showed that Nfe2 gene transduction into 4T1.0 cells enhanced their capability to form intraosseous tumours, whereas Nfe2 shRNA treatment reduced tumour formation of 4T1.3 clone, as well as another mouse TNBC-derived TS/A.3 clone with an augmented intraosseous tumour formation ability. They propose that Nfe2 can potentially contribute to breast cancer cell survival in bone microenvironment under hypoxic condition.

The manuscript has been improved by incorporating some of the suggested experiments. Although not all the suggestions have been incorporated, I believe that the data are more consistent and better supported than the previous version.

In summary the addition of new experiments and clarifications in the text have improved the final manuscript. In my opinion, it is almost acceptable for publication.

We express our sincere gratitude to the thoughtful and favorable comments.

Reviewer 2 Report

xcv

Authors extensively revised the manuscript and the manuscript is significantly improved. However, the reply for the Comment 2 is not satisfactory yet.

1. In supplementary Figure S1 (a), authors present the histological pictures of 4T1 tumor in bone with HE staining. However, histology with TRAP staining for osteoclasts is not presented in the revised manuscript. Authors have cited a paper published from their group (Ref. 10) to justify their proposal that 4T1.3 tumors grow in bone via osteoclast-independent mechanism. However, in the study reported in Ref. 10, osteoclasts were enumerated as CD51+CD45+CD11b+Ly6C+ cells, but not TRAP+ cells. Further, osteoclast number was counted at day 7, at which stage osteoclastic bone resorption may not be necessary yet. It has been widely-accepted that bone resorption by osteoclasts is a critical requisite for cancer to progress in bone and that the cells of osteoclast lineage are the only cell capable to resorb bone. One thus wonders how 4T1.3 grows in bone, a hard calcified tissue, without the assistance of osteoclasts. In this context, determination of the effects of increased Nfe2 in 4T1.3 cells on osteoclastogenesis and bone resorption should be intriguing.
2. In bone metastasis, OPG and RANKL are provided by osteocytes, osteoblasts and stromal cells and bind to RANK on osteoclasts. It is therefore suggested that authors determine if 4T1.3 cells in which Nfe2 expression is increased produce a factor that stimulates RANKL expression and inhibits OPG expression in osteocytes, osteoblasts and stromal cells. Have authors determined the effects of the conditioned medium harvested from 4T1.3 cultures on the expression of OPG/RANKL and RANK in these cells compared with that from 4T1.0?
3. In supplementary Figure S1 (c), authors present the result of RANK/RANKL/OPG expression in 4T1.0 and 4T1.3. The data do not properly answer the question. To determine the effects of Nfe2, overexpression or knockdown experiment is suggested. Further, the expression of RANK/RANKL/OPG needs to be examined in 4T1.0 and 4T1.3 cells growing in bone, not in culture.

Author Response

Authors extensively revised the manuscript and the manuscript is significantly improved. However, the reply for the Comment 2 is not satisfactory yet. 

  Q1) In supplementary Figure S1 (a), authors present the histological pictures of 4T1 tumor in bone with HE staining. However, histology with TRAP staining for osteoclasts is not presented in the revised manuscript. Authors have cited a paper published from their group (Ref. 10) to justify their proposal that 4T1.3 tumors grow in bone via osteoclast-independent mechanism. However, in the study reported in Ref. 10, osteoclasts were enumerated as CD51+CD45+CD11b+Ly6C+ cells, but not TRAP+ cells. Further, osteoclast number was counted at day 7, at which stage osteoclastic bone resorption may not be necessary yet. It has been widely-accepted that bone resorption by osteoclasts is a critical requisite for cancer to progress in bone and that the cells of osteoclast lineage are the only cell capable to resorb bone. One thus wonders how 4T1.3 grows in bone, a hard calcified tissue, without the assistance of osteoclasts. In this context, determination of the effects of increased Nfe2 in 4T1.3 cells on osteoclastogenesis and bone resorption should be intriguing.

Q3) In supplementary Figure S1 (c), authors present the result of RANK/RANKL/OPG expression in 4T1.0 and 4T1.3. The data do not properly answer the question. To determine the effects of Nfe2, overexpression or knockdown experiment is suggested. Further, the expression of RANK/RANKL/OPG needs to be examined in 4T1.0 and 4T1.3 cells growing in bone, not in culture.

In response to the comment, we first determined osteoclast activation by the immunohistochemical analysis using anti-CD51, which can be a reliable marker of osteoclasts [1]. As we described in the third paragraph of page 2 and showed in the new Figure 1e, intraosseous injection of either 4T1.0 or 4T1.3 increased intraosseous CD51-positive osteoclast numbers to similar extents, compared with untreated mice. We admitted that the description in the previous manuscript may cause impression that osteoclast activation was not involved in this bone metastasis model, but what we would like to claim is that osteoclast activation was not augmented upon intraosseous injection of 4T1.3, compared with that of 4T1.0. Thus, exaggerated osteoclast activation alone cannot account for enhanced intraosseous tumor formation upon 4T1.3 injection, compared with 4T.0 injection. Accordingly, we added the legend to the new Figure 1e and 1f and modified the Materials and Method section (third paragraph of page 12).

[1] Hamzei M, Ventriglia G, Hagnia M, et al. Osteoclast stimulating and differentiating factors in human cholesteatoma. Laryngoscope. 2003;113(3):436-442.

In response to the recommendation to examine the effects of Nfe2, we determined the expression of RANK/RANKL/OPG in Nfe2-expressing or Nfe2-shRNA-treated cells. As we demonstrated in Figure 3d and 4e and described in the second paragraph of page 4 and the first paragraph of page 6, Nfe2 gene transduction or Nfe2 shRNA treatment had few effects on the in vitro expression of these molecules.

Additionally, we agreed with the recommendation that the expression of RANK/RANKL/OPG needs to be examined in 4T1.0 and 4T1.3 cells growing in bone, not in culture. Hence, as we demonstrated in the new Figure 1f and described in the third paragraph of page 2 that any significant differences were observed between 4T1.0-injected and 4T1.3-injected mice.

   Q2) In bone metastasis, OPG and RANKL are provided by osteocytes, osteoblasts and stromal cells and bind to RANK on osteoclasts. It is therefore suggested that authors determine if 4T1.3 cells in which Nfe2 expression is increased produce a factor that stimulates RANKL expression and inhibits OPG expression in osteocytes, osteoblasts and stromal cells. Have authors determined the effects of the conditioned medium harvested from 4T1.3 cultures on the expression of OPG/RANKL and RANK in these cells compared with that from 4T1.0?

Nfe2 expression was expressed selectively in 4T1.3 cells grown in a bone cavity but not under in vitro conditions. Thus, it may be highly possible that 4T1.3 cells cannot in vitro produce some bioactive substances in an Nfe2-dependent manner. Thus, we dared not determine the effect of conditioned medium harvested from cultured 4T1.0 or 4T1.3 cells.

This manuscript is a resubmission of an earlier submission. The following is a list of the peer review reports and author responses from that submission.

Round 1

Reviewer 1 Report

This MS is well done.

For triple negative breast cancer wie need als much als data availale to geht visions about new Therapeutin options.

this MS contributes to such a mandatory need

Reviewer 2 Report

Manuscript CANCERS: 887829

            The manuscript entitled” Involvement of a transcription factor, Nfe2, in breast      cancer metastasis to bone” by Dr. Di Zhang et al., present some data using a 4T1.3 tumour cell clone with a high capacity to metastasize to bone after orthotopic injection, derived from a murine TNBC cell line, 4T1.0.

They describe some aspects of the molecular mechanism underlying a high growth ability of 4T1.3 in a bone cavity. They performed a gene expression analysis and they detected enhanced Nfe2 mRNA expression in 4T1.3 grown in a bone cavity, compared with in vitro culture conditions.

They showed that Nfe2 gene transduction into 4T1.0 cells enhanced their capability to form intraosseous tumours, whereas Nfe2 shRNA treatment reduced tumour formation of 4T1.3 clone, as well as another mouse TNBC-derived TS/A.3 clone with an augmented intraosseous tumour formation ability.

They propose that Nfe2 can potentially contribute to breast cancer cell survival in bone microenvironment under hypoxic condition.

            The manuscript shows some interesting data; however, I consider that still have some missing data to support the working hypothesis.

 The rationale of this work is correctly exposed perhaps until the Figure 4 , from here I detect a little  “jump”  to the Wnt data (Figure 5), that in some way is disconnected with the previous data.

       Q1)   The main problem is about the conclusion after Figure 4,…The authors say…..

These observations would indicate that Nfe2 could enhance the growth of breast cancer cells in a bone cavity where hypoxia predominates”.

I strongly suggest adding some data to support this hypothesis.;  for instance the measured of HIF1a, HIF2, NrF2 NFkB or Myc ,  from  4T1.3 versus 4T1.0 cells; or 4T1.0 cells versus 4T1.0 cells after Nfe2 gene transduction…etc would give us  a more accurate hypothesis  to support the conclusions.

            Q2) The data from Figure 5, support the conclusion that 4T1.0-Nfe2 cells   upregulated some elements of the Wnt pathway, when compared with 4T1.0 cells. I agree with the authors the data. This does not imply that this upregulation is linked with the survival enhancement in hypoxic conditions. In my opinion this interesting observation is not linked with the previous data.

 I suggest adding some data perhaps after inhibition of Wnt pathway, o after reducing expression of LEF1…to reinforce the final hypothesis that “Wnt upregulation is linked with the capacity of some breast cancer cells to survive in bone microenvironment under hypoxic condition”.

Minor point.

Almost all data are based on the mRNA quantification of several genes implicated and described. However, in some cases (in my personal experience) … some genes upregulated at mRNA levels are not linked directly with the increasing amount of protein, sometimes does not happens.  Perhaps, as Suppl. Figure, I would like to suggest adding the data protein levels of Nfe2 in

4T1.0-Nfe2 cells  

4T1.0 cells versus

4T1.3 cells

4T1.3 cells after interference with Nfe2- shRNA

In summary the manuscript contains new and very interesting data. I think that with some additions and some clarifications, the final manuscript would be improved. In my opinion almost ready to be acceptable after adding some data.

Reviewer 3 Report

/c/

In this study, authors showed that the expression of the transcription factor Nfe2 was increased in breast cancer colonized in bone and that Nfe2 increases the progression of 4T1 mouse breast cancer in bone with increased expression of Wnt-related molecules including Axin2, Lbh, and Lef1, while Nfe2 had no effects on the progression 4T1 cancer at primary site. Authors also demonstrated that Nfe2 changes cell survival and anchorage-independent growth of breast cancer cells in culture under hypoxic, but not normoxic, condition. Since bone microenvironment is hypoxic, authors propose that Nfe2 regulates breast cancer growth selectively in bone with activation of Wnt-related molecules.

Major comments:

  1. The manuscript provides unique information that the expression of the transcription factor Nfe2 is increased in breast cancer colonized in bone, but not at primary site, regulating breast cancer colonization in bone, but not at primary m.f.p.
  2. However, the manuscript is phenomenological and lacks the data of biological mechanism by which Nfe2 regulates breast cancer behaviors selectively in bone. It is widely known that bone resorption is tightly associated with breast cancer metastasis to bone. Accordingly, the effects of Nfe2 on RANKL/RANK/OPG expression, osteoclast differentiation and bone destruction need to be examined by H.E. and TRAP staining and bone histomorphometry. Further, the effects of Nfe2 on VEGF expression and angiogenesis, which are critical for cancer progression and metastasis, also need to be investigated by CD31 staining and capillary tube formation by HUBEC. Immunohistochemistry of 4T1 tumor in bone by Ki67 staining to determine 4T1 cell proliferation is also suggested.
  3. Along this line, why shNfe2 4T1.3 progression was decreased in bone, but not changed in m.f.p. in Figure 3? What biological events were involved here?
  4. To support the notion that hypoxia of bone microenvironment up-regulates Nfe2 function, it is mandatory to determine HIF-1α expression in 4T1 breast cancer and interactions between HIF-1α and Nfe2.
  5. Are any of bone metastasis-related genes expressed in 4T1 breast tumor transcriptionally regulated by Nfe2, except for Ccl4?
  6. Fig 1c, have authors determined Nfe2 expression in 4T1.3 at primary site and in lung compared to that in bone? Nfe2 expression in 4T1.3 cells in vitro is not an appropriate control for comparison.
  7. Fig 2d, have authors determined survival of 4T1.3-bearing mice compared to that of 4T1.0-bearing mice?

Minor comments:

  1. Description of the techniques to identify candidate transcription factors in Results 2.1, Figure 1 legend and Method is repetitive, please rewrite.
  2. Fig 4 is difficult to understand, please show the data of cell number and survival rate separately. It is unclear how survival rate was determined. The culture medium was not changed during experiments? Some experimental controls are missing in Figure 4 (a)-(f), namely, anchorage-independent in normoxia, and anchorage-dependent in hypoxia.
  3. Figure 4 (b), and (d), the statistical difference in cell number appears very marginal, is this biologically significant?
  4. Authors, please discuss by what mechanism Wnt-signal activation leads to promotion of breast cancer colonization in bone.
  5. This is just personal interest of this reviewer, does bone, especially osteoclasts, in Nfe2 KO and TG mice show any phenotype?